# Coagulation Dysfunctions in Non-Alcoholic Fatty Liver Disease—Oxidative Stress and Inflammation Relevance

**DOI:** 10.3390/medicina59091614

**Published:** 2023-09-07

**Authors:** Madalina Andreea Robea, Ioana-Miruna Balmus, Irina Girleanu, Laura Huiban, Cristina Muzica, Alin Ciobica, Carol Stanciu, Carmen Diana Cimpoesu, Anca Trifan

**Affiliations:** 1CENEMED Platform for Interdisciplinary Research, “Grigore T. Popa” University of Medicine and Pharmacy, 700115 Iasi, Romania; madalina.robea11@gmail.com (M.A.R.); balmus.ioanamiruna@yahoo.com (I.-M.B.); carmen.cimpoesu@umfiasi.ro (C.D.C.); 2Department of Exact Sciences and Natural Sciences, Institute of Interdisciplinary Research, “Alexandru Ioan Cuza” University of Iasi, Alexandru Lapusneanu Street, No. 26, 700057 Iasi, Romania; 3Department of Gastroenterology, “Grigore T. Popa” University of Medicine and Pharmacy, 700115 Iasi, Romania; gilda_iri25@yahoo.com (I.G.); huiban.laura@yahoo.com (L.H.); lungu.christina@yahoo.com (C.M.); ancatrifan@yahoo.com (A.T.); 4Institute of Gastroenterology and Hepatology, “St. Spiridon” University Hospital, 700111 Iasi, Romania; 5Department of Biology, Faculty of Biology, “Alexandru Ioan Cuza” University, Carol I Avenue, No. 20A, 700505 Iasi, Romania; 6Centre of Biomedical Research, Romanian Academy, Carol I Avenue, No. 8, 700506 Iasi, Romania; stanciucarol@yahoo.com; 7Academy of Romanian Scientists, Splaiul Independentei nr. 54, Sector 5, 050094 Bucuresti, Romania; 8Department of Emergency Medicine, Emergency County Hospital “Sf. Spiridon”, 700111 Iasi, Romania; 9Faculty of Medicine, University of Medicine and Pharmacy “Grigore T. Popa” Iasi, Blvd. Independentei 1, 700111 Iasi, Romania

**Keywords:** hepatic steatosis, steatohepatitis, reactive oxygen species, lipid peroxidation, mitochondria, endoplasmic reticulum, macrophages, foamy cells, adipokines, thrombosis, hypercoagulability

## Abstract

Non-alcoholic fatty liver disease (NAFLD) is one of the most common liver diseases. Its incidence is progressively rising and it is possibly becoming a worldwide epidemic. NAFLD encompasses a spectrum of diseases accounting for the chronic accumulation of fat within the hepatocytes due to various causes, excluding excessive alcohol consumption. In this study, we aimed to focus on finding evidence regarding the implications of oxidative stress and inflammatory processes that form the multifaceted pathophysiological tableau in relation to thrombotic events that co-occur in NAFLD and associated chronic liver diseases. Recent evidence on the pathophysiology of NAFLD suggests that a complex pattern of multidirectional components, such as prooxidative, proinflammatory, and prothrombotic components, better explains the multiple factors that promote the mechanisms underlying the fatty acid excess and subsequent processes. As there is extensive evidence on the multi-component nature of NAFLD pathophysiology, further studies could address the complex interactions that underlie the development and progression of the disease. Therefore, this study aimed to describe possible pathophysiological mechanisms connecting the molecular impairments with the various clinical manifestations, focusing especially on the interactions among oxidative stress, inflammation, and coagulation dysfunctions. Thus, we described the possible bidirectional modulation among coagulation homeostasis, oxidative stress, and inflammation that occurs in the various stages of NAFLD.

## 1. Introduction

Currently, non-alcoholic fatty liver disease (NAFLD) is considered one of the most common chronic liver diseases, with rising epidemiological evidence suggesting that it is a possible source of a modern non-communicable disease epidemic [1,2,3]. In 2020, the estimation of the global prevalence of the disease was almost 2 billion cases, making NAFLD the most prevalent disease in human history [3]. The latest global estimation of NAFLD prevalence [4] revealed an overall estimation of almost 4.7 cases per 100 individuals, with a significant difference based on gender (7.08 cases per 100 males and 2.96 cases per 100 females). Despite this significant prevalence in males (age ≤ 50–60 years) compared to females, a high incidence of NAFLD among postmenopausal women (aged 50–60 years and older) was recorded, suggesting that hormone replacement therapy may exhibit prosteatotic properties [5].

Generally, NAFLD is characterized by the abnormal accumulation of fat in the liver, in the absence of significant alcohol use, viral hepatitis, or prosteatotic medication, with or without liver function disturbance [6]. Some authors consider that NAFLD is the hepatic manifestation of the metabolic syndrome, due to the increased incidence of the occurrence of associated risk factors, such as obesity (especially visceral fat), type 2 diabetes, dyslipidemia, and hypertension [7,8,9]. Since the incipient stages are asymptomatic, NAFLD is usually diagnosed during routine abdominal imaging or blood tests [10,11]. Despite the fact that, in some cases, NAFLD presents benign stability (non-alcoholic fatty liver), it was demonstrated that hepatic steatosis could further progress and lead to hepatic inflammation (non-alcoholic steatohepatitis (NASH)), liver fibrosis, or cirrhosis). Furthermore, up to 2.6% of NAFLD cases, as well as 3.8% to 4.6% of the NAFLD-related cirrhosis cases, were found to progress to hepatocellular carcinoma (HCC) [2,12].

Despite the long history of research in NAFLD pathophysiology, there are still aspects that need further attention, especially regarding the mechanisms through which NAFLD progresses to NASH and cirrhosis. In this context, it was shown that NAFLD is an umbrella term for complex and multifaceted hepatic steatosis, which is currently known to include multiple components [13]. Oxidative stress and inflammation were found to influence the progression and aggravation of NAFLD significantly [14,15]. Moreover, increasing evidence regarding the occurrence of thrombotic events in both simple and aggravated NAFLD suggests the possible prothrombotic condition of hepatic steatosis [16,17]. Despite this, the implications of oxidative stress and inflammatory processes in the prothrombotic state of NAFLD are still under discussion.

Thus, in this study, we aimed to focus on finding evidence regarding the implications of oxidative stress and inflammatory processes that form the multifaceted pathophysiological tableau in relation to thrombotic events that co-occur in NAFLD and associated chronic liver diseases.

## 2. Non-Alcoholic Fatty Liver Disease

Non-alcoholic fatty liver disease (NAFLD) is characterized by the accumulation of fat, which affects more than 5% of hepatocytes. In contrast to the broader spectrum of metabolic-associated fatty liver disease, which offer a more inclusive diagnosis to any pathology associated with fatty liver, NAFLD excludes viral hepatitis, alcohol consumption, and medications that predispose patients to hepatic steatosis [18,19]. Nevertheless, exposure to alcohol consumption and prosteatotic medication could further increase the predisposition and severity of liver damage, due to steatosis and/or hepatic inflammation [18]. In addition, it was recently suggested that NAFLD may not be a component of metabolic syndrome, but rather a key factor in the pathophysiology of that syndrome, with hepatic damage being one of the targets in this multiorgan chronic disease [20]. This distinct NAFLD pathogenesis could be explained by the occurrence of NAFLD in the absence of obesity (lean NAFLD) [21]. Lean NAFLD patients often show signs of impaired liver function (as suggested by the abnormal liver transaminase activity) and comorbid type 2 diabetes mellitus, dyslipidemia, or hypertension, in the absence of obesity [22].

Hepatic steatosis may be indicated by typical biochemical profiles of the blood, such as dyslipidemia, hypercholesterolemia, or hyperglycemia [22]. In some cases, signs of hepatic steatosis could be observed during abdominal examination (Table 1) [23,24]. 

The NAFLD activity score (NAS) is calculated as the sum of the scores of steatosis, lobular inflammation, and hepatocyte ballooning. NAS ≤ 3 correlates with a “not NASH” diagnosis and NAS ≥ 5 correlates with a “definite NASH” diagnosis [25]. The fibrosis-4 score (FIB-4) is based on age, aspartate aminotransferase (AST), alanine aminotransferase (ALT), and platelets. FIB-4 < 1.45 refers to a “no advanced fibrosis” and FIB-4 > 3.25 refers to “advanced fibrosis” [30].

The histological features of NAFLD mainly include macrovesicular hepatic steatosis, but microvesicular hepatic steatosis was also reported in almost 10% of the cases [33,34]. No specific mechanisms were described to differentiate between macro- and microvesicular types, and it is thought that they reflect the different phases of fat accumulation within the hepatocytes, starting with endoplasmic reticulum membranes and ending with a single prominent lipid vesicle that pushes the nucleus and the cytoplasm to peripheral positions [35,36]. However, in NASH, microvesicular steatosis is accompanied by megamitochondria, or giant mitochondria, which highly correlates with the effects of oxidative stress in hepatic tissues, mainly due to increased lipid peroxidation [37].

In the more aggressive form of NASH, hepatic fat accumulation is accompanied by hepatic inflammation that could further lead to tissue damage (cirrhosis). Some NASH cases also affect the spleen and the circulatory system, causing splenomegaly and blood vessel enlargement due to the overexpression of proinflammatory and proangiogenic factors [27,28,29]. In contrast to hepatic steatosis, NASH has distinct and specific histological features. Accordingly, NASH was classified in necro-inflammatory grades as a function of hepatic damage severity (from mild to severe) and in stages based on damage localization (perisinusoidal, portal, bridging fibrosis, and cirrhosis, in a cumulative manner) [37]. The inflammatory processes undergoing NASH pathophysiology lead to the cellular hallmark of hepatocytic ballooning with or without Mallory–Denk bodies [38]. During the inflammatory processes caused by fat accumulation, hepatocyte cytoskeleton filaments are severely altered, while they are dispersed to the periphery of the cells, alongside the rarified cytoplasm, or form highly eosinophilic Mallory–Denk bodies with a central position [37,39]. The mild intralobular inflammation and the rare portal inflammation are characterized by inflammatory cell infiltrates, scattered lobular microgranulomas, lipogranulomas, and acidophil bodies, as a result of apoptotic, necrotic, and fibrotic events [37].

The key differences between NAFLD and alcoholic fatty liver disease (AFLD) are the patterns of hepatic fatty degeneration and inflammatory cell infiltration [40,41]. While the clinical differentiation is usually caused by the history of alcohol consumption, both NAFLD and AFLD have shared pathogenic mechanisms that are closely related to oxidative homeostasis and the inflammatory response [42]. However, although ethanol hepatic metabolism and toxicity were not correlated to oxidative stress [43], the overproduction of oxidative ethanol metabolites, as potent oxidative stress promoters, received full credit for oxidative damage [44].

In the more severe cases, cirrhotic NASH could further progress to a particular form of HCC [45]. The reason why some NAFLD cases persist in a benign and non-progressive manner, while others fast and aggressively progress to NASH, cirrhosis, and/or HCC is not currently fully understood, yet it is thought that the primary site of this multiorgan pathology remains the liver, while the most frequent complication causing increased mortality is cardiovascular disease [46]. Recent studies suggested that the fast progression of NAFLD may be the result of genetic predisposition, as the presence of some single-nucleotide polymorphisms was found to potentiate the risk for advanced fibrosis in a prooxidative and proinflammatory cellular environment [47,48,49].

As NAFLD pathogenesis was found to be much more complex than initially thought, considering various risk factors, the need for an etiological theory that could bind together all the pathogenic components was pressing. Thus, the ”two-hit” theory proposed a pathogenic mechanism tableau that partly explains the still-controverted NAFLD pathogenesis [50,51]. However, the two hits were quickly proved to be multiple [51,52].

In the category of first hit for NAFLD are included a series of risk factors, including overnutrition, obesity, dietary composition, and sedentary lifestyles, along with the well-known factor of insulin resistance (IR) [53]. There are several NAFLD cases that are not linked to IR as the main factor in their development [51]. Overnutrition and dietary composition patterns are usually a mark of obesity, which is more pronounced when it is associated with an inactive lifestyle. For example, a recent meta-analysis showed that the prevalence of NAFLD in overweight and obese people is almost 70%, with the highest prevalence in America [54]. On the other hand, China has recorded a 30% prevalence rate for NAFLD prevalence, but its rate among obese children/adolescents ranged from 57.6% to 71.7% for different regions of the country [55]. 

Closely related to obesity, the lack of physical activity has been correlated with the prevalence of NAFLD [56,57]. It seems that technological advances have led to a reduction in healthy habits and the promotion of sitting activities. Studies from 2020 and 2023, evaluating the South Korean population, proved that there is a positive correlation between NAFLD and time spent sitting. Moreover, it has emerged that the risk of developing NAFLD increases in direct proportion to the time spent in inactivity or sitting [58,59]. 

As mentioned above, dietary composition also contributes to the occurrence, progression, and support of NAFLD-specific molecular changes. In the general context of NAFLD, it is thought that a high-fat diet heavy in trans fats, saturated fats, cholesterol, and fructose-containing beverages may increase visceral fat, enhance lipid accumulation in the liver, and accelerate the progression to steatohepatitis and fibrosis [60,61]. Although obese individuals manifest a higher incidence of NAFLD, this trend has started for non-obese individuals as well, according to recent data [62,63].

Normally, insulin encourages the uptake of glucose and the synthesis of fatty acids from nonlipid molecules, which is known as lipogenesis. Insulin stimulates glycolysis by activating glucokinase, together with hepatic pyruvate kinase, acetyl-CoA carboxylase, fatty acid synthase, stearoyl-CoA desaturase, sterol regulatory element-binding protein 1c (SREBP-1c), and Spot 14, which leads to FA synthesis [64]. In general, the resistance of insulin decreases the hormone’s capacity to diminish glucose production; however, at the same time, it can support lipogenesis [65]. Some hypotheses indicate that β-oxidation can enhance IR, but coexisting with this process, reduced levels of reactive oxygen species (ROS) might provoke insulin signaling [65].

In some cases, the first hit of NAFLD is not represented by IR [66,67]. The existence of several genetic variants that appear as single-nucleotide polymorphisms (SNPs) suggests multiple genetic pathways for NAFLD. The most well-known SNP for NAFLD is rs738409 C>G, which encodes the protein variant of patatin-like phospholipase domain-containing 3 (PNPLA3) [68,69]. This promotes the fast and aggressive progression of NAFLD to NASH via altered lipid catabolism within the hepatocytes containing endoplasmic-reticulum membranes and lipid-vesicles membranes that present the PNPLA3 I148M variant [70]. The effect of genetic variants in the PNPLA3 gene on NAFLD can differ among ethnic groups; for instance, the Chinese people having the lowest effect of genetic variants, compared to Indians or Malays [71]. A recent genome-wide association study also identified several new predisposition locations for genes implicated in steroid dehydrogenase oxidoreductase activity modulation (17-beta hydroxysteroid dehydrogenase 13—HSD17B13) and endoplasmic-reticulum membrane-modulated lipid metabolic processes (Transmembrane 6 superfamily Member 2—TM6SF2) [72,73]. HSD17B13 is a protein found on the surface of the lipid droplets within hepatocytes, the loss of which was associated with a reduction in NAFLD progression to NASH or cirrhosis [74]. TM6SF2 was recently reported as an independent risk factor for NAFLD and used as a predictor for fibrosis progression and HCC occurrence [75].

## 3. Oxidative Stress and Inflammation in Non-Alcoholic Fatty Liver Disease

Together with the strong lipid-profile changes leading to hepatic steatosis, the repercussions on the mitochondrial function, the immune system (innate and adaptive immunity), and the intestinal microbiota were extensively described [17]. In this way, oxidative stress and inflammatory processes were found to be unbalanced components of a multisystemic interplay.

Oxidative stress is caused by an imbalance between the bioavailability of reactive oxygen species (ROS) and the cellular antioxidant system. Consequently, significant deficiencies in cellular activities eventually lead to cell death [76]. ROS play physiological roles at low concentrations; at high concentrations, they could react with cellular components. Moreover, ROS excess can contribute to DNA damage, chromosome instability, aberrant gene expression, and genetic mutations that eventually promote structural and functional liver impairments. On the other hand, ROS are directly implicated in protein and lipid oxidative damages, due to their increased reactivity with molecular constituents of the cells, such as structural proteins and cell membranes. Furthermore, many enzymes and membrane proteins also interact with ROS, leading to alterations or loss of functions. The main endogenous generators of ROS are mitochondria, phagocyte cells, peroxisomes, cytochrome P450 enzymes, NADPH oxidases, inducible nitric oxide production (iNOS), and heme proteins that are involved in liver functions [77,78,79]. Thus, due to the vital processes attributed to it, the liver seems responsible for most of the metabolic pathways implicated in detoxification, leading to a high generation of oxidative products [80]. In addition, hepatocytes are responsible for storing vitamins (A, B, D, E, and K), glycogen, and minerals, such as iron and copper, which are involved in ROS-producing reactions. It was shown that disturbances in the metabolism of vitamin A were positively correlated with NAFLD progression caused by vitamin accumulation in liver cells [81]. The vitamin B group was noted as having multiple implications in hepatocytes, including an increase in redox status, mitochondrial metabolism, and ER stress [82]. Regarding the role of vitamins D and E in NAFLD, the scientific data reported mixed results. Both vitamins are known for their antioxidant and anti-inflammatory properties [82,83]. As for vitamin K in the NAFLD context, existing knowledge points to its participation in lipid metabolism and in the coagulation cascade [82,84].

Notably, all chronic liver diseases share as a common hallmark—the presence of a highly oxidative environment and inflammation—which sustains hepatocellular damage and contributes to the progression of fibrosis, cirrhosis, and, eventually, HCC. In the case of NAFLD and NASH, ROS and reactive nitrogen species (RNS) result from mitochondrial, endoplasmic reticulum, and peroxisome activities (Table 2) [65].

In addition to oxidative stress, another important pathophysiological component of NAFLD is the sustained inflammatory process. Despite the fact that macro- and microvesicular hepatic steatosis cause a mild inflammatory response, it was shown that oxidative stress also contributes to NAFLD inflammation, which eventually leads to the progression of simple NAFLD to NASH. In this context, the next step in progression is promoted by fibrogenic mechanisms, while hepatic stellate cells (HSCs), portal fibroblasts, and myofibroblasts are activated by proinflammatory cytokines and adipokines [85,86]. Eventually, the advanced fibrosis causes permanent scarring of the liver tissues and cirrhotic processes [31]. In the initiation and progression of NAFLD, lymphocytes play a critical role by triggering the generation of IL-17, Th17-derived cytokine, and TNFα, which are linked with increased liver inflammation. Elevated levels of these molecules was found in the NAFLD livers of patients with an increased body mass index [90]. In addition, in NAFLD patients with obesity, IR acts as a promoter of oxidative stress and is also involved in macrophage recruitment, due to proinflammatory cytokines signaling [91,92]. Furthermore, it was demonstrated that macrophage recruitment is directly correlated to lipotoxicity and to adipocytes apoptosis [93].

IR is considered as the major contributor for NAFLD development and progression. IR is sometimes interpreted as the first hit, while the second hit could be oxidative stress [51,94]. Based on subsequent fat accumulation as a result of a passage mechanism in preventing liver toxicity, bidirectional signaling causes hepatic de novo lipogenesis and adipose tissue lipolysis, leading to an increased influx of lipids into liver tissue [95]. Consequently, a vicious cycle between insulin signaling and lipid mobilization fuels sustained lipotoxicity and glucotoxicity. In this proinflammatory and prooxidative environment, mitochondrial dysfunction (as a result of excessive hepatocytic lipid metabolites), the presence of adipokines, the secretion of proinflammatory cytokines, and endoplasmic reticulum stress all contribute to the second hit by causing ROS overproduction, which eventually depletes antioxidant reserves [51]. 

Mitochondria are responsible for generating energy, primarily through β-oxidation and the tricarboxylic acid (TCA) cycle [77]. In a healthy liver, as a result of fatty-acid oxidation and ATP, mitochondria supply glucose production in hypoglycemic conditions. Mitochondria from NAFLD patients are usually fragmented, with increased calcium storage and excessive mitochondrial ROS products that lead to c-Jun N-terminal kinase (JNK) activation as part of the mitogen-activated kinase cascade in the metabolic stress adaptation of hepatocytes [96]. Consequently, this produces similar mitochondrial effects, creating a feed-forward cycle [96,97]. In addition, JNK activation in the stages of steatohepatitis contributes to the increase in SAB protein expression found on the outer mitochondrial membrane. Their combination is linked with defective mitochondrial electron transport chains, the discharge of oxidative stress products, and, as a final effect, hepatocyte death [97,98].

Decreased fatty-acid oxidation caused by this impairment of the mitochondrial function is hypothesized to induce free fatty-acid accumulation in hepatocytes, while simultaneously impairing insulin signaling. In the hepatocyte cytosol with fatty-acid catabolism, free fatty acids are converted to fatty acyl-CoA, which then enters mitochondria via carnitine palmitoyl-transferase 1 (CPT1). β-oxidation decomposes free fatty acids to acetyl-CoA, which can be decomposed into carbon dioxide and water through the TCA cycle. In addition to these two mechanisms that are associated with NASH progression from NAFLD, another key factor is located in the mitochondria’s inner membrane, due to the disruption of ATP synthesis [65]. This happens as a consequence of the high influx of free fatty acids coupled with TCA’s defective function, which results in increased ROS generation [99].

Sometimes, due to their multiple implications for cell function, mitochondria are capable of experiencing several processes, including biogenesis, fission, fusions, or mitophagy [100]. Each process is triggered as an adjustment to the metabolic requirements. For instance, the growth and split of existing mitochondria are usually known as biogenesis, while fission and fusion are processes implicated in the separation and, respectively, the merger of mitochondria [101,102]. Most of the time, obesity is mentioned as a risk factor for the onset of NAFLD [103,104,105]. Due to IR, the excess of energy and increased lipolysis sustain the appearance of dysfunctional adipose tissue in the liver and other body tissues, as observed through the free movement of fatty acid [105]. It was shown that there is a strong correlation between IR, decreased mitochondrial biogenesis, and elevated levels in the mitochondrial fission process [106]. Additionally, mitochondrial biogenesis can be perturbed by the decrease of the peroxisome proliferator-activated receptor γ coactivator-1α (PGC1α), which contributes to gluconeogenesis, fatty acid oxidation, and lipid transport gene expression [107]. In addition, PGC1α exhibited lower expression in the liver of sedentary and obese individual [108]. Related to the involvement of PGC1α in HCC, it was recently shown that PGC1α regulates the activity of peroxisome proliferator-activated receptor (PPAR) α, one of the controlling agents in carcinogenesis [109], and further, its overexpression inhibits tumor cell migration and invasion [110].

On the opposite side of biogenesis, there is the fission–fusion interaction that is constantly adapting, depending on the mitochondrial and cell environments. Fission of mitochondria can occur as a result of oxidative stress, leading to the formation of damaged mitochondria from healthy ones [111]. A key factor in the mitochondrial fission process is the activity of dynamin-related protein 1 (DRP1), which acts strictly on apoptosis by mediating the process of fission [112]. It was suggested that the loss of liver DRP1 may enhance the inflammatory response [112], and its phosphorylation, as a consequence of ROS products, encourages the fission process [113]. Moreover, findings from 2022 highlighted that excessive mitochondrial fission may make a certain contribution to the progression of NAFLD, while the inhibition of this process was correlated to a decrease in hepatic injuries and endoplasmic reticulum (ER) stress [106].

On both mitochondrial membranes, specific proteins, such as: mitofusin 1 (MFN1) and mitofusin 2 (MFN2), control the fusion process on the outer membrane and optical atrophy 1 (OPA1) on the inner membrane [87,114]. Decreases in these protein levels contribute to a reduction in mitochondrial respiration and mitochondrial membrane potential [100,115]. The fusion process can be promoted and potentiated as an outcome of the presence of oxidative stress in response to changes in cell dynamics, with significant impact in mitochondrial functions, calcium homeostasis, and ATP production [116]. In addition, alterations of mitochondrial fusion can lead to a NASH-like phenotype and HCC’s particular characteristics [112]. It was proven that NAFLD progression, inflammation, and hyperglycemia can be amplified by the deletion of hepatocyte-specific MFN2, which is known to be involved in mitochondrial fragmentation [117,118].

The involvement of interleukin 6 (IL-6) in the pathogenesis of NAFLD is usually seen as a possible therapeutic target, as it participates in numerous cellular events. In this case, the blockage of the IL-6 signal leads to the activation of the mitochondrial fusion gen [119]. IL-6 is implicated in inflammation, redox processes, and the promotion of IR. The promotion of IR is indicated by scientific data that support the capacity of IL-6 to amplify IR in the hepatic cells [120]. In addition, the quantification of IL-6, together with interleukin 8 and tumor necrosis factor α (TNF α), was proposed to predict NASH progression to HCC [121]. 

Together with these processes, in order to prevent mitochondrial failure, specific mechanisms enable damaged mitochondria to be repaired via a mitochondrial unfolded protein response or disrupted via autophagy (known as mitophagy) [87]. For instance, the failure of mitophagy was observed in the liver of several NASH individuals [122], and its reduction has been directly linked to the severity of the diseas [123].

ER is another organelle implicated in ROS generation through the oxidoreductase and NADPH oxidase catalytic processes. ER is known to be engaged in several cellular processes, especially protein synthesis and transportation, as well as calcium deposition. The involvement of ER in NAFLD is represented by its direct implication in lipid synthesis in the hepatocytes [124,125]. Frequently, ER stress is mentioned in the context of NAFLD, considering its role in sustaining IR in fat metabolism [124,126,127]. ER stress and the concomitant unfolded protein response (UPR) contribute to its pathogenesis, primarily due to the high ER content in the liver [128]. Activation of the UPR is dependent on the arousal of three transmembrane endoplasmic reticulum stress sensors—inositol-requiring enzyme 1 (IRE1), PKR [double-stranded RNA-activated protein kinase]-like ER kinase (PERK), and activating transcription factor 6 (ATF6). Its activity is inhibited by the ER chaperone GRP78/Binding immunoglobulin protein (BiP) [128,129,130]. In normal conditions, the BiP expression level is low. Thus, in ER stress, unfolded protein starts to accumulate in the ER and, further, the GRP78 molecule binds to it concomitantly with ER stress response signals modulated by the changes in the XBP-1 marker [131]. Simultaneously, if CHOP (a transcription factor of C/EBP) expression is intensely increased, the occurrence of apoptosis is definitely inevitable [126,130].

Furthermore, in a state of oxidative stress, mitochondria and ER are connected via mitochondrial-associated membranes (MAMs) that intervene in the exchange of Ca^2+^, lipids, and ROS molecules [125]. Ca^2+^ transport is facilitated by MAMs from the ER to the mitochondria, and its dysfunctional transport is associated with mitochondrial perturbations, ER stress, IR, and lipid accumulation [132,133,134]. It was indicated that the accumulation of fat in NAFLD livers with MAM alterations leads to a disrupted Ca^2+^ balance, which promotes the progression of NAFLD marked by the presence of oxidative stress, inflammation, and apoptosis [132,133].

The third source of intracellular ROS and RNS products is peroxisomes. During the process of β-oxidation, acetyl-CoA molecules are formed from FA through the intervention of acyl-CoA oxidase, which is not metabolized in the mitochondria [78]. In addition, β-oxidation can start after peroxisome stimulation, caused by the activation of the peroxisome proliferator-activated receptor (PPAR) α, a receptor linked to the AOX enzyme [77,78]. 

However, recent evidence showed that oxidative stress and inflammatory processes could hardly be described separately in the NAFLD context, due to the multiple cross-signals that contribute to the bidirectional potentiation of both mechanisms, which viciously support each other.

## 4. Coagulation Dysfunctions in Non-Alcoholic Fatty Liver Disease

The homeostasis mechanism, through which blood coagulation is assured, resides in the perfect balance between procoagulant and anticoagulant factors within the vascular system. The main purpose of blood coagulation remains the prevention of blood constituents scattering throughout the tissues and extracellular environments when blood-vessel-wall injuries occur, in which case the formation of a thick pellet of fibrin may rescue the vessel and promote its repair. The physiological mechanisms of coagulation were previously extensively described [135,136]. However, the way in which coagulation dysfunctions interact with NAFLD pathophysiology is not entirely understood, despite the fact that some authors consider NAFLD to be a prototype disorder for coagulation abnormalities [137].

The most common coagulation dysfunctions are clotting factor deficiencies, hypercoagulable states, and deep-venous thrombosis, which all occur during NAFLD progression [138]. Frequent thrombotic events and modified coagulation time are reported in chronic liver diseases, due to the liver’s inability to secrete the coagulation factors [139,140,141]. As the impaired mechanism of coagulation in cirrhosis is rather simple, a complex interplay between impaired liver functions and prothrombotic states, fueled by the altered metabolic context, was previously described in NAFLD (Figure 1).

Many studies regarding the hypercoagulation state in NAFLD reported that arterial and venous-thrombosis risks observed in patients are frequently associated with IR [17]. In this way, fibrinolytic processes and endothelial and platelet dysfunction mechanisms were closely related to the mechanisms underlying IR. The implication of adipokines in IR was previously demonstrated [142,143,144], and the presence of visceral fat tissues was directly correlated with impaired clotting dynamics in NAFLD [145,146].

On the other hand, it was shown that a hypercoagulation state could induce the progression of NAFLD-associated hepatic injury. In this way, abnormal circulation, hepatic thrombosis, and ischemia could all be implicated in the hepatocellular destruction that eventually promotes the progression of NASH to cirrhosis. The development of NASH and fibrosis was previously obtained by coagulation cascade activation. In addition, NASH-specific inflammation was observed when excessive hepatic thrombin and fibrinogen syntheses were not modulated by lipid accumulation [17].

Atherosclerosis and venous thromboembolisms were also described in NAFLD, with higher frequency [147,148,149]. Zeina et al. [149] provided evidence to demonstrate that the incidence of thromboembolic events in NAFLD is not strictly correlated with the presence of cirrhosis, as previously thought. In this context, they argued that coagulation defects within NAFLD pathomechanisms are the result of hepatocellular destruction, and they suggested that other molecular processes may be involved. In addition, they implied that neither inflammation nor coagulation defects are mandatory for the occurrence of NAFLD pathomechanisms [149].

In this context, platelet functionality and endothelial functions were found to be altered by increased ROS production [49]. It was recently proposed that altered platelet functions could promote NAFLD development. Yin et al. [150] described the molecular pathway through which the platelet-activating factor, as increased platelet recruitment, was frequently associated with NAFLD in the context of IR and altered aminotransferase activity. Furthermore, they discussed the implication of the platelet-activating factor in NAFLD, while considering that cyclooxygenase inhibitors could successfully alleviate cytokine-induced inflammation, and concluded that the platelet-activating factor is capable of promoting NAFLD-associated mechanisms that modulate oxidative stress, inflammatory responses, or IR. Moreover, Nasiri-Ansari et al. [151]. described the implication of liver sinusoidal endothelial cells in NAFLD progression, according to their potential to regulate the inflammatory process, hepatic stellate activation, and augmented vascular resistance, leading to endothelial tension and altered microcirculation.

Nevertheless, excessive ROS production and accumulation cause aberrant interactions with essential molecules that are implicated in structural and functional homeostasis. In this way, the ROS capacity to oxidize low-density lipoproteins (LDL) was described as being implicated in the transformation of macrophages into foamy cells that further promote atherogenesis [152,153]. In NAFLD, foamy Kupffer cells were seen to undergo this action against oxidized lipoproteins while performing other cellular signaling, such as hepatocyte interaction via cytokines, chemokines, and enzymes, as a response to ROS and nitric oxide, immune response initiation, and debris scavenging [154].

Aside from the acquired coagulation defects due to liver dysfunction, recent studies showed that certain genetic variations in individuals could provide sufficient predisposition to NAFLD and impairment in the early steps of fibrogenesis [155]. However, the genetic predisposition to procoagulant imbalance was not associated with liver fibrosis in NAFLD [155]. Nevertheless, some of the genetic predisposition variants associated with NAFLD were seen in the encoding factors of several key genes that regulate oxidative homeostasis. Carpino et al. [156] showed that PNPLA3 variant carriers exhibit increased oxidative stress and portal fibrogenesis, thus suggesting a possible yet unsolved association with oxidative dyshomeostasis and coagulation dysfunction. Additionally, the deletion of the TM6SF2 variant was linked with mitochondrial and ER stress as an outcome of the organelle’s impaired function [157]. Consequently, the current state of knowledge suggests a bidirectional modulation among coagulation homeostasis, oxidative stress, and inflammation, which occurs in the various stages of NAFLD, in light of specific risk factors.

## 5. Possible Interaction Mechanisms among Coagulation Dyshomeostasis, Oxidative Stress, and Inflammation

Considering that NAFLD frequently coexists with other components of the metabolic syndrome, including diabetes, hypertension, hyperlipidemia, and obesity, which are common risk factors for cardiovascular disease, assessing the impact of liver disease on its own is rather difficult [158,159]. Despite this, recent reports showed an increased incidence of venous and pulmonary thromboembolism and splanchnic venous thrombosis, suggesting that the procoagulant state is strongly correlated with prothrombotic events seen in NAFLD patients [160,161]. In this way, endothelial dysfunction, platelet function impairments, coagulation cascade imbalance, and fibrinolytic activity could further contribute to hepatic injury, as seen in NAFLD—“parenchymal extinction” followed by fibrous septa [162,163]. This process is characterized by the appearance of microthrombi that interfere with blood flow and lead to congestion, local ischemia, and hepatocyte apoptosis as a final result (Figure 2) [17]. As it was previously shown that significant oxidative stress is produced by the impairment of the metabolism of hepatic cells, ROS accumulation could further contribute to mitochondrial and endoplasmic reticulum function disruption, thus contributing to more metabolic imbalance [127]. Additionally, it was found that ROS can act within the red blood cells, leading to an imbalance of redox status and, moreover, these cells can be supplemented with more ROS present in the extracellular matrix that came from other cells releases [164,165]. Therefore, the elevated ROS level in red blood cells is seen as a potential agent for the initiation of venous thrombosis [164]. A significant association between oxidative stress and coagulation status in NAFLD can be established, due to the complexity of interactions among hepatocytes, red blood cells, platelets, and other molecules involved in this process. For instance, Ogresta et al. [17]. explained the multifactorial interplay among genetic predisposition, low-grade chronic inflammation, adipokines, oxidative stress, platelet abnormalities, and coagulation impairment to describe the increased mortality occurring in NAFLD patients, due to acquired co-morbid cardiovascular diseases. Furthermore, they suggested that the efficiency of statins in NASH and atherosclerotic patients could be due to their anti-inflammatory, antioxidant, and antifibrotic activity [17]. In this context, it was shown that the mechanisms through which ROS could contribute to the development of NAFLD and the progression to NASH, and that HCC impairs endothelial function, coagulation homeostasis, and platelet functionality when ROS are over-produced and accumulated by blood vessel walls and hepatocytes [49,166].

Except for the Von Willebrand factor (VWF), the pro- and anticoagulant factors are exclusively produced by the liver’s parenchymal cells (I, II, V, VII, VIII, IX, XI, XII, and XIII) [167,168]. For instance, in cirrhotic patients, the complex formed between FVIII and VWF was found at significantly increased levels [169]. Given the two roles of VWF in hemostasis, increasing platelet adherence to thrombogenic surfaces, as well as platelet-to-platelet cohesion during thrombus formation and VWF;s role as a plasmatic carrier for FVIII, the impairment of VWF functions is often due to its cleaving enzyme ADAMTS13 malfunctioning, leading to platelet microthrombi formation, sinusoidal microcirculatory abnormalities, and, eventually, hepatic damage [168]. As a predictable boomerang, if enzyme ADAMTS13 levels diminish as a result of hepatic apoptosis, and if HSCs are the main producers of this enzyme, VWF levels increase concomitantly with the rise in FVIII and coagulation promotion (Figure 3) [170,171]. Lombardi et al. [171]. suggested that chronic liver diseases could be associated with reduced ADAMTS13 activity, and defective ADAMTS13 synthesis could be a consequence of liver inflammation, as seen in NASH.

The severity of the main factor responsible for liver-related outcomes—liver fibrosis—was linked to higher circulating levels of FVIII, which were accompanied by a decrease in protein C (PC) activity, resulting in an increased FVIII/PC ratio [16,161,172]. PC, the natural anticoagulant from blood, acts as an anti-inflammatory and anti-apoptotic participant, as well as having an impact on gene modulation in endothelial cells and leukocytes [173]. Additionally, PC regulates factor X and prothrombin through the activated FVIII and factor V (FVIIIa, FVa) [174].

Although the exact root cause of the increase in FVIII levels in liver diseases is not known, there are several hypotheses regarding why FVIII levels changed [175,176,177]. The release of cytokine from necrotic cirrhotic liver tissue might result in elevated FVIII synthesis [178]. Even more likely, the overproduction of cytokines and iNOS by hepatic Kupffer cells (KCs) and monocytes and lymphocytes stimulates a proinflammatory environment [179]. ROS produced by KCs stimulate HSCs by increasing the proliferation and production of extracellular matrices, which contributes to fibrosis and cirrhosis (Figure 3) [180]. Oxidative stress, in combination with inflammation, generates localized or zonal necrosis, hepatocyte death, and architectural disturbance [181].

Extrahepatic FVIII synthesis sites (the spleen, the kidneys, the lungs, and the endothelium) are also thought to be sources of enhanced FVIII synthesis [167]. Potential additional causes of elevated FVIII levels include abnormal endothelial protein production, FVIII misfolding that interferes with secretion, impaired protein catabolism, and factors regulating FVIII, such as increased hepatic VWF biosynthesis, decreased low-density lipoprotein receptor-related protein expression, or protein C deficiency [167]. For instance, Poothong and colleagues demonstrated that ER stress can be activated by increased FVIII expression, which ends with amyloid-like fibrils; a process that might be reversed through the restoration of glucose metabolism [182]. In NAFLD, the coagulation cascade leads to thrombin production after being activated by the VWF and plasminogen activator inhibitor type I (PAI-1), which stimulate platelet hyperactivity mediated by the proteinase activated receptor 1–4 (PAR1–4) [183]. A correlation between PAI-1, endoplasmic reticulum stress, and NAFLD progression to NASH was previously shown in murine models of NAFLD/NASH [184,185].

Thus, it was shown that oxidative stress and inflammation are strongly linked with the major alterations that occur in the hepatic tissues as a consequence of NAFLD pathogenesis, including increased ROS production, which lead to fibrosis, cirrhosis, and, finally, hepatocellular carcinoma [32]. In addition, aging significantly contributes to several altered functions of each liver cell type (hepatocytes, liver sinusoidal endothelial cells, HSCs, and KCs), leading to possible vulnerability to oxidative stress and further promoting NAFLD-related pathomechanisms [169]. Furthermore, in addition to the well-known consequences of ROS occurrence (changes in molecular structures and DNA damage), the presence of senescence-associated secretory phenotype (SASP) cells was shown to promote the recruitment of inflammatory cells ands to intervene in blood clotting by activating platelets, leading to thrombotic events [169,186,187].

## 6. Conclusions

Non-alcoholic fatty liver disease (NAFLD) is currently considered one of the main causes of chronic liver disease, with a multifactorial pathogenesis and various manifestations, from simple hepatic steatosis to hepatocellular destruction, predisposing to coagulopathies and hepatic malignancies. Taking into consideration the interactions among the hepatocyte’s activity in lipotoxicity, glucotoxicity, and coagulation, the current state of knowledge suggests a bidirectional modulation among coagulation homeostasis, oxidative stress, and inflammation occurring in the various stages of NAFLD.

## Figures and Tables

**Figure 1 medicina-59-01614-f001:**
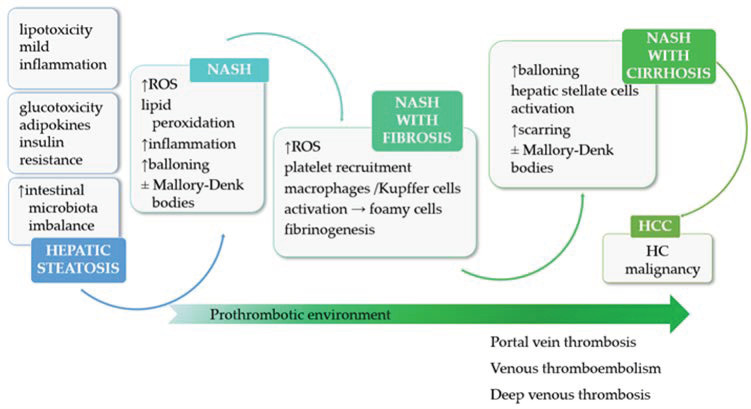
The stages of NAFLD pathophysiology and its progression in HCC, and subsequent prooxidative, proinflammatory and prothrombotic events. (NASH: nonalcoholic steatohepatitis, ROS: reactive oxygen species, HC: hepatocellular, HCC: hepatocellular carcinoma, ↑––increased; ↓––decreased).

**Figure 2 medicina-59-01614-f002:**
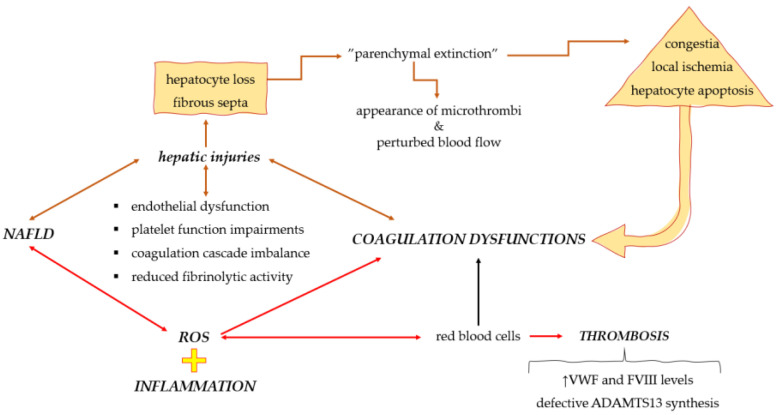
Schematic representation of the interaction established between NAFLD and coagulation dysfunctions. (FVIII: factor VIII; ROS: reactive oxygen species; VWF: Von Willebrand factor).

**Figure 3 medicina-59-01614-f003:**
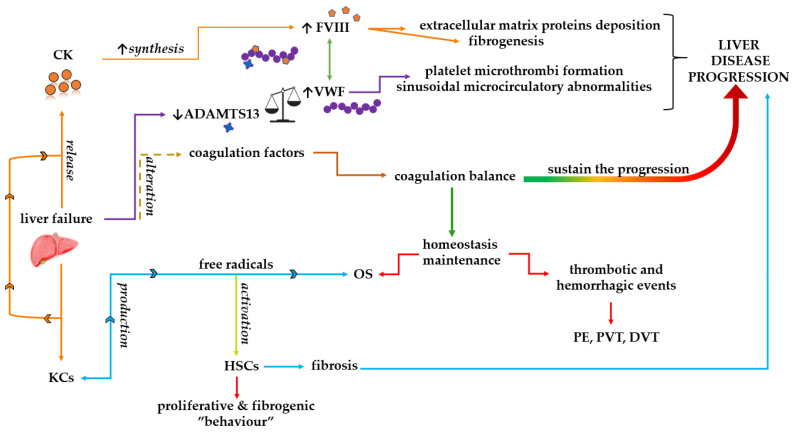
Schematic representation of several alterations that occur in the coagulation cascade associated with NAFLD and oxidative stress. The failure of liver leads to the release of more CK in response to necrotic tissue, which triggers the synthesis of FVIII. When FVIII, which is known to be implicated in thrombin generation, exceeds normal levels, it provokes the appearance of extracellular matrix protein deposition and fibrosis. In the same context of liver injury, a low level of the enzyme ADAMTS13 generates the formation of platelet microthrombi and impacts the sinusoidal microcirculation. In addition, changes in the activity of coagulation factors can damage the coagulation balance. Coagulation balance refers to the interaction between the procoagulant pathways specific for clot formation and those mechanisms included under the name of the fibrinolysis system. In addition, HSCs are activated in the presence of free radicals, initiating the proliferative and fibrogenic “behavior” that sustains the progression of liver disease. Consequently, oxidative stress is able to perturb the homeostasis maintenance by inducing the appearance of thrombotic and hemorrhagic events. (CK: cytokines; DVT: deep vein thrombosis; FVIII: factor VIII; HSCs: hepatic stellate cells; KCs: Kupffer cells; OS: oxidative stress; PE: pulmonary embolism; PVT: portal vein thrombosis; VWF: Von Willebrand factor).

**Table 1 medicina-59-01614-t001:** Pathophysiological traits of chronic hepatic diseases.

Chronic Hepatic Diseases	Pathophysiological Traits	References
Hepatic steatosis	liver imaging: hyperechogenic liver parenchyma; dyslipidemia, hypercholesterolemia, or hyperglycemia.	[22,24]
NAFLD	>5% hepatocytes with fat accumulation; abnormal liver transaminases activity, dyslipidemia, insulin resistance (IR); mild liver inflammation; absence of significant alcohol consumption, viral hepatitis and medications that predispose to hepatic steatosis; NAFLD activity score (NAS) ≤ 3; Fibrosis-4 score (FIB-4) < 1.45.	[18,19,22,25,26]
NASH	hepatic fat accumulation; hepatic inflammation (intralobular and portal); endothelial tissue changes; ±hepatic necrosis; ±hepatic cirrhosis; ±hepatic fibrosis; absence of hepatotoxic medications, autoimmune liver diseases; NAS ≥ 5.	[25,27,28,29,30]
Cirrhosis	hepatic fat accumulation; hepatic inflammation; hepatic scarring; parenchymal extinction; FIB-4 > 3.25.	[26,30,31,32]

**Table 2 medicina-59-01614-t002:** Histological and molecular changes in NAFLD.

Chronic Hepatic Diseases	Histological Changes	Oxidative Stress	Inflammation	References
NAFLD	macrovesicular hepatosteatosis (single prominent lipid vesicle, nucleus with peripheric positioning); fat accumulation in hepatocytic endoplasmic reticulum membranes; ultrastructural changes in megamitochondria classified as elongated, irregular, and spheroidal giant mitochondria resulted from the disorganization of cristae, matrix granules, and filamentous intramitochondrial crystalline inclusions.	↑ ROS production; IR-driven oxidative stress; mitochondria, endoplasmic reticulum, and peroxisomes dysfunction.	↓ fatty acid oxidation; hepatic stellate cells (HSCs), portal fibroblasts, and myofibroblast activation; defective mitochondrial electron transport chain; proinflammatory cytokines modulated macrophage recruitment.	[35,36,51,85,86,87,88]
NASH	microvesicular steatosis; megamitochondria usually seen within hepatocytes with microvesicular steatosis; hepatocytic ballooning; ±Mallory–Denk bodies.	↑ ROS production; ↑ lipid peroxidation; mitochondria, endoplasmic reticulum, and peroxisome dysfunction.	inflammatory cells infiltrate; scattered lobular microgranulomas; lipogranulomas; acidophilic bodies.	[37,38,87,89]

↑––increased; ↓––decreased.

## Data Availability

Not applicable.

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
