# Peer review of "Coagulation Dysfunctions in Non-Alcoholic Fatty Liver Disease—Oxidative Stress and Inflammation Relevance"

_medicina, 2023, doi:10.3390/medicina59091614_

Round 1

Reviewer 1 Report

This manuscript entitled "Coagulation Dysfunctions in Non-Alcoholic Fatty Liver Disease - Oxidative Stress and Inflammation Relevance" is very important because NAFLD is one of the most common chronic liver diseases. NAFLD pathogenesis and metabolic disruption are still not clear enough. Also, to date, there are no FDA-approved drug therapies for NAFLD/NASH. The term MAFLD  has been recently proposed to more accurately reflect the disease’s heterogeneity and pathogenesis. The authors aimed to describe possible pathophysiological mechanisms, especially focusing on the interaction between oxidative stress, inflammation, and coagulation dysfunctions in the various stages of NAFLD. There is strong evidence that mitochondrial dysfunction plays a significant role in the development and progression of NAFLD.

Minor comments:

In paragraph about mitochondrial dysfunction and oxidative stress should be added in detail mitochondrial dynamics (the fusion and fission of mitochondria, mitofusin-1 and mitofusin-2 ), Biogenesis,  and Mitophagy in NAFLD. Also, in paragraph about ER stress should be added mitochondria-associated membranes (MAMs) because disruption of MAM integrity, miscommunication directly or indirectly causes imbalances in Ca2+ homeostasis and increases ERS and oxidative stress.

Author Response

As the reviewer kindly indicated, we added the mentioned suggestions in the manuscript, and due to this intervention, the quality of the present work increased considerably. Thus, here are some of our additions related to mitochondrial dynamics: ”Sometimes, due to its multiple implications in cells function, mitochondria are capable to experience several processes as: biogenesis, fission and fusions or mitophagy [65]. Each process is triggered as an adjustment for the metabolic requirements. For instance, the growth and the split of existing mitochondria is usually known as biogenesis, while fission and fusion are processes implicated in the separation, and respectively the merge of mitochondria [66], [67]. Most of the times, obesity is mentioned as a risk factor for NAFLD onset [68]–[70]. Due to the IR, the excess of energy and increased lipolysis sustain the appearance of dysfunctional adipose tissue into the liver and other body tissues observed through the free fatty acid’s movement [70]. It was shown that there is a strong correlation between IR, decreased mitochondrial biogenesis, and elevated levels in mitochondrial fission process [71]. Additionally, mitochondrial biogenesis can be perturbed by the decrease of the peroxisome proliferator-activated receptor γ coactivator-1α (PGC1α) that contributes to the gluconeogenesis, fatty acid oxidation, lipid transport genes expression [72]. Also, the PGC1α exhibited lower expression in the liver of sedentary and obese individuals [73]. Related to the its involvement in HCC, it was recently showed that PGC1α regulates the activity of peroxisome proliferator-activated receptor (PPAR) α, one of the controlling agents in carcinogenesis [74], and further, its overexpression in the inhibition of tumor cell migration and invasion [75].

On the opposite part of biogenesis, there is the fission-fusion interaction that is constantly adapting depending the mitochondrial and cell environment. Fission of mitochondria can occur as a result of oxidative stress leading to the formation of damaged mitochondria from healthy ones [76]. A key factor in the mitochondrial fission process is the activity of dynamin-related protein 1 (DRP1), acting strictly on apoptosis by mediating the process of fission [77]. It was suggested that the loss of liver DRP1 may conduct to an enhance of the inflammatory response [77], and its phosphorylation, as a consequence of ROS products, encourages the fission process [78]. Moreover, findings from 2022 were highlighting that excessive mitochondrial fission may have a certain contribution to the NAFLD progression, while inhibition of this process was corelated to a decrease in hepatic injuries and endoplasmic reticulum (ER) stress [71].

On both mitochondrial membranes, specific proteins control the fusion process, such as: mitofusin 1 (MFN1) and mitofusin 2 (MFN2) on the outer membrane and optical atrophy 1 (OPA1) on the inner membrane [79], [80]. Decreases of these protein levels contribute to a reduction of mitochondrial respiration and mitochondrial membrane potential [65], [81]. The fusion process can be promoted and potentiated as an outcome of oxidative stress presence in response to the cell dynamics changes with significant impact on mitochondrial functions, calcium homeostasis, and ATP production [82]. Also, the alterations of mitochondrial fusion can conduct to NASH-like phenotype and HCC particular characteristics [77]. It was proven that NAFLD progression, inflammation, and hyperglycemia can be amplified by the deletion of MFN2 hepatocyte-specific; known to be involved in the mitochondrial fragmentation [83], [84].

The involvement of interleukin 6 (IL-6) in the pathogenesis of NAFLD is usually seen as a possible therapeutic target since it participates in numerous cellular events. In this case, when IL-6 signal is blocked it leads to the activation of mitochondrial fusion gene [85]. IL-6 is implicated in inflammation, redox processes, and in the promotion of IR; the last being sustained by the scientific data which supports the capacity of IL-6 to amplify the IR in the hepatic cells [86]. In addition, the quantification of IL-6 besides interleukin 8, and tumor necrosis factor α (TNF α) was proposed to predict NASH progression to HCC [87].

Together with these processes, in order to prevent mitochondrial failure, specific mechanisms enable it to either repair damaged mitochondria via mitochondrial unfolded protein response or disrupt it via autophagy; known as mitophagy [80]. For instance, the failure of mitophagy was observed in the liver of several NASH individuals [88], and its reduction has been directly linked to disease severity [89].” and ”Furthermore, in a state of oxidative stress, mitochondria and ER are connected via mitochondrial-associated membranes (MAMs) that intervene in the exchange of Ca2+, lipids, and ROS molecules [84]. Ca2+ transport is facilitated by MAMs from ER to mitochondria, and its dysfunctional transport is associated with mitochondrial perturbations, ER stress, insulin resistance, and lipid accumulation [91]–[93]. It was indicated that the ac-cumulation of fat in NAFLD livers with MAM alterations conducts to a disrupted Ca2+ balance, that promotes the progression of NAFLD marked by the presence of oxidative stress, inflammation, and apoptosis [91], [92].”.

Thank you!

Kind regards,

The authors.

Reviewer 2 Report

This manuscript aims to address coagulation dysfunctions in Non-alcoholic fatty liver disease (NAFLD), with a focus on inflammation and oxidative stress. Unfortunately, the article does not effectively address this dysfunction. While some parts are interesting, crucial information about the relationship between coagulation, oxidative stress, and inflammation is omitted. The article instead contains an extensive introduction to the pathophysiological mechanisms of NAFLD and NASH. Additionally, certain sentences do not align logically with the proposed content. Here are some examples of necessary improvements:

The graphical abstract should be revised to accurately represent the review. Furthermore the information in it must be checked. For instance, chronic or persistent inflammation can result in tissue damage and the initiation of fibrogenesis. Thus, it is not solely linked to coagulation dysfunction and oxidative stress. Another illustration is the association of insulin resistance with inflammation beyond oxidative stress. While oxidative stress is one facet of inflammation contributing to insulin resistance, there are additional inflammatory mechanisms in action.

In Table 1, the authors should remove "suggestive percutaneous liver biopsy" as it is not a pathophysiological trait of NASH but a diagnostic technique. Irrelevant information should be eliminated, and more precise descriptions of cited scores are required. It is unclear why NASH and cirrhosis are included in this table, as the review's focus is on oxidative stress and inflammation in thrombotic events related to NAFLD.

Table 2 could be improved by consistently expressing the characteristics of the condition or indicating that they may be present (e.g., "megamitochondria"). The presence of NASH features in this table is unclear and should be addressed.

In line 177, replace "ROS-proliferating" with "ROS-producing". Also, clarify that the cited vitamins are primarily associated with reducing ROS reactions.

Line 199: The assertion that insulin resistance is the first hit of NAFLD should be either eliminated or better explained, considering its absence in some cases of NAFLD.

Figure 1: While NASH can be considered a stage of NAFLD, the same cannot be said for HCC. Revise the legend to accurately reflect this distinction.

Lines 374-377: The authors mention "significant oxidative stress produced by parenchymal cells metabolism impairment," but there is no previous mention of this in the article. Either correct the sentence or provide relevant context.

Line 377: The statement "a significant association between oxidative stress and coagulation status in NAFLD was established" requires further explanation or revision, because it was not.

Figure 2's coagulation aspect needs clarification. The phrase "coagulation balance" should be defined; does it refer to balanced coagulation factor synthesis contributing to homeostasis, leading to thrombotic and hemorrhagic events? Additionally, the role of alteration factors in liver disease progression should be clarified and the figure revised accordingly.

Overall, the manuscript requires significant revisions to address the aforementioned issues and enhance the clarity and coherence of its content.

Author Response

  1. The graphical abstract should be revised to accurately represent the review. Furthermore, the information in it must be checked. For instance, chronic or persistent inflammation can result in tissue damage and the initiation of fibrogenesis. Thus, it is not solely linked to coagulation dysfunction and oxidative stress. Another illustration is the association of insulin resistance with inflammation beyond oxidative stress. While oxidative stress is one facet of inflammation contributing to insulin resistance, there are additional inflammatory mechanisms in action.

            According to the reviewer’s suggestion, we modified the graphical abstract paying attention to the interactions established between the oxidative stress, inflammation, and coagulation dysfunctions that are correlated through multiple common consequences as: insulin resistance, fibrogenesis, and impaired hemostasis.

  1. In Table 1, the authors should remove "suggestive percutaneous liver biopsy" as it is not a pathophysiological trait of NASH but a diagnostic technique. Irrelevant information should be eliminated, and more precise descriptions of cited scores are required. It is unclear why NASH and cirrhosis are included in this table, as the review's focus is on oxidative stress and inflammation in thrombotic events related to NAFLD.

We removed the "suggestive percutaneous liver biopsy" from the Table 1 and we modified the information added in the table as the reviewer kindly suggested. Regarding the presence of NASH and cirrhosis in this table, it were added in order to have a perspective view since there are multiple reports which highlights the fact that some individuals with NAFLD can develop NASH and further cirrhosis. In addition, we believed that adding this table will help readers to obtain a better understanding about the pathophysiological traits of the hepatic diseases mentioned along the manuscript. Also, the descriptions of the scores was added as: ” NAS (NAFLD activity score) calculated as the sum scores of steatosis, lobular inflammation and hepatocyte ballooning; NAS≤ 3 correlated with ”not NASH” diagnosis and NAS≥ 5 with ”definite NASH” [25]. FIB-4 (Fibrosis-4 score) based on age, aspartate aminotransferase (AST), alanine aminotransferase (ALT), and platelet; FIB-4< 1.45 refers to a ”no advanced fibrosis” and FIB-4 > 3.25 for ”advanced fibrosis” [30].”

  1. Table 2 could be improved by consistently expressing the characteristics of the condition or indicating that they may be present (e.g., "megamitochondria"). The presence of NASH features in this table is unclear and should be addressed.

The Table 2 was improved with add-ins as the reviewer kindly specified; for instance, the histological changes column was completed for both NAFLD (”Ultrastructural changes of megamitochondria classified in elongated, irregular, and spheroidal giant mitochondria resulted from the disorganization of cristae, matrix granules and filamentous intramitochondrial crystalline inclusions.”) and NASH: ”megamitochondria usually seen within hepatocytes with microvesicular steatosis”. Concerning the presence of NASH in this table, it was added due to high prevalence of NAFLD progression to NASH, an advanced form of NAFLD characterized by severe changes at histological and molecular levels.

  1. In line 177, replace "ROS-proliferating" with "ROS-producing". Also, clarify that the cited vitamins are primarily associated with reducing ROS reactions.

            We replaced in the manuscript "ROS-proliferating" with "ROS-producing" as it was suggested, and we kindly clarified the implications of the mentioned vitamins in the ROS reactions as: ”It was showed that disturbances in the metabolism of vitamin A were positively correlated with NAFLD progression caused by the vitamin accumulation in liver cells [55]. The vitamin B group was mentioned to have multiple implications in hepatocytes including an increase in redox status, mitochondrial metabolism, and ER stress [56]. Regarding vitamin D and E, the scientific data reports mixed results concerning their role in NAFLD. Both vitamins are known for its antioxidant and anti-inflammatory properties [56], [57]. As for vitamin K in NAFLD context, existing knowledge points out its participation in lipid metabolism and in the coagulation cascade [56], [58].”

  1. Line 199: The assertion that insulin resistance is the first hit of NAFLD should be either eliminated or better explained, considering its absence in some cases of NAFLD.

            We modified the phrase where the assertion that insulin resistance is the first hit of NAFLD in: ”IR is considered as the major contributor for NAFLD development and progression; sometimes being interpreted as the first hit, while the second one could be the oxidative stress.”, and we still kept the formulation ”first hit” since there are numerous reports using it even if now it is discussed about a multiple-hit theory instead of ”two-hit theory” due to its complexity (for instance: Dowman et al., 2009 - doi.org/10.1093/qjmed/hcp158; Paschos et al., 2009 - www.ncbi.nlm.nih.gov/pmc/articles/PMC2683460/; Buzzetti et al., 2016 - doi.org/10.1016/j.metabol.2015.12.012; Chen et al., 2017 - doi.org/10.1186/s12944-017-0572-9; Fang et al., 2018 - 10.3748/wjg.v24.i27.2974).

  1. Figure 1: While NASH can be considered a stage of NAFLD, the same cannot be said for HCC. Revise the legend to accurately reflect this distinction.

            We thank to the reviewer for this observation and we corrected the title of Figure 1 as: ”The stages of NAFLD pathophysiology and its progression in HCC, and subsequent pro-oxidative, pro-inflammatory and pro-thrombotic events (NASH: Nonalcoholic steatohepatitis, ROS: reactive oxygen species, HC: hepatocellular, HCC: hepatocellular carcinoma)”.

  1. Lines 374-377: The authors mention "significant oxidative stress produced by parenchymal cells metabolism impairment," but there is no previous mention of this in the article. Either correct the sentence or provide relevant context.

We rephrased the sentence from: ”As it was previously showed that significant oxidative stress is produced by parenchymal cells metabolism impairment, ROS accumulation could further contribute to hepatocytes’ mitochondria and endoplasmic reticulum functions disruptions, thus to more metabolic imbalance” to ”As it was previously showed that significant oxidative stress is produced by hepatic cells metabolism impairment, ROS accumulation could further contribute to mitochondria and endoplasmic reticulum functions disruptions, thus to more metabolic imbalance.”

  1. Line 377: The statement "a significant association between oxidative stress and coagulation status in NAFLD was established" requires further explanation or revision, because it was not.

In order to improve and clarify the statement mentioned above, we kindly added several phrases as: ”Additionally, it was found that ROS can act within the red blood cell conducting to an imbalance of redox status and, moreover, these can be supplemented with more ROS present in the extracellular matrix that came from other cells release [143], [144]. So, the elevated ROS level in the red blood cells is seen as a potential agent for the initiation of venous thrombosis [143]. Therefore, a significant association between oxidative stress and coagulation status in NAFLD can be established due to its complexity of interactions built between hepatocytes, red blood cells, platelet, and other molecules involved in this process .”

  1. Figure 2's coagulation aspect needs clarification. The phrase "coagulation balance" should be defined; does it refer to balanced coagulation factor synthesis contributing to homeostasis, leading to thrombotic and haemorrhagic events? Additionally, the role of alteration factors in liver disease progression should be clarified and the figure revised accordingly.

            We added more information related to the Figure 2's coagulation aspect as the reviewer kindly indicated such as: ”Figure 2. Schematic representation of several alterations which occurs in the coagulation cascade associated with NAFLD and oxidative stress. The failure of liver leads to the release of more CK in response to necrotic tissue that triggers the synthesis of FVIII. When FVIII, known to be implicated in the thrombin generation, exceeds the normal levels, provoke the appearance of extracellular matrix proteins deposition and fibrosis. In the same context of liver injury, low level of enzyme ADAMTS13 generates the formation of platelet microthrombi and impacts the sinusoidal microcirculation. Also, the changes occurred in the activity of coagulation factors can damage the coagulation balance. Coagulation balance refers to the interaction between the procoagulant pathways specific for clot formation and those mechanisms included under the name of fibrinolysis system. In addition, HSCs are activated in the presence of free radicals initiating the proliferative, and fibrogenic ”behaviour” that sustain the progression of liver disease. Consequently, oxidative stress is able to perturb the homeostasis maintenance by inducing the appearance of thrombotic and hemorrhagic events. (CK: cytokines; DVT: deep vein thrombosis; FVIII: factor VIII; HSCs: hepatic stellate cells; KCs: Kupffer cells; OS: oxidative stress; PE: pulmonary embolism; PVT: portal vein thrombosis; VWF: Von Willebrand factor).”

Thank you!

Kind regards,

The authors.

Round 2

Reviewer 2 Report

Despite the authors' responses to the majority of the comments, the review remains confusing. Many parts reference hypotheses, and it's essential to explicitly state these instances in the article.

Here are some suggested modifications:

·        The graphical abstract can still be improved. For example, consider placing the new addition depicting thrombotic and hemorrhagic events inducing ROS accumulation on the left side. This placement would enhance the connection with oxidative stress. The same applies to fatty acid oxidation. Is it driven by inflammation or oxidative stress? Is the current placement optimal?

·        As noted in the initial revision, this review remains unclear, as various chapters provide scattered information that lacks cohesion.

·        Chapter 3 should elucidate the relationship between oxidative stress and inflammation and how they interconnect. Readers anticipate this order based on the title. Additionally, restructure and rewrite this chapter. Elements somewhat linked to both topics should be placed in the NAFLD pathophysiology chapter.

·        The "two-hit" model is a hypothesis that explains NAFLD progression. The first "hit" involves fat accumulation in liver cells due to increased fatty acid uptake and decreased lipid oxidation, leading to hepatic steatosis. The second "hit" includes factors triggering inflammation, oxidative stress, and cellular damage, which may progress to NASH and fibrosis. This model emphasizes lipid accumulation and inflammation's pivotal roles. The subsequent chapter discusses a multiple hit model, yet it's slightly confusing to insert insulin resistance solely as a first hit. Consider being more inclusive about possible first hit components, such as sedentary lifestyle, high-fat diet, and obesity, as mentioned in the cited paper: "Hepatic accumulation of lipids secondary to sedentary lifestyle, high fat diet, obesity, and insulin resistance, acts as the first hit, sensitizing the liver to further insults acting as a ‘second hit’" (Buzzetti et al., 2016). Highlight the fact that while insulin resistance is vital, it's not the exclusive "first hit" for NAFLD. Mention that there's literature showcasing NAFLD cases without insulin resistance (see https://www.mdpi.com/1422-0067/17/5/633).

·        Despite the figure legend, Figure 2 still requires modification. Replace "disturbance" with "sustain the progression."

·        Adding a figure illustrating the relationship between coagulation factors and NAFLD will significantly enhance the paper, as it aligns with one of the central aims of the article.

Overall, the review's structure and content need further refinement to enhance clarity and coherence.

Author Response

  1. The graphical abstract can still be improved. For example, consider placing the new addition depicting thrombotic and hemorrhagic events inducing ROS accumulation on the left side. This placement would enhance the connection with oxidative stress. The same applies to fatty acid oxidation. Is it driven by inflammation or oxidative stress? Is the current placement optimal?

We thank to the reviewer for its suggestions; it brought a new face to the graphical abstract and it seems much more suitable for this review.

  1. As noted in the initial revision, this review remains unclear, as various chapters provide scattered information that lacks cohesion.

The manuscript has been read and rephrased in order to obtain a new and approachable perspective for the readers.

  1. Chapter 3 should elucidate the relationship between oxidative stress and inflammation and how they interconnect. Readers anticipate this order based on the title. Additionally, restructure and rewrite this chapter. Elements somewhat linked to both topics should be placed in the NAFLD pathophysiology chapter.

The entire chapter 3 has been restructured according to the reviewer’s indication and we focused on presenting the interaction between oxidative stress and inflammation.

  1. The "two-hit" model is a hypothesis that explains NAFLD progression. The first "hit" involves fat accumulation in liver cells due to increased fatty acid uptake and decreased lipid oxidation, leading to hepatic steatosis. The second "hit" includes factors triggering inflammation, oxidative stress, and cellular damage, which may progress to NASH and fibrosis. This model emphasizes lipid accumulation and inflammation's pivotal roles. The subsequent chapter discusses a multiple hit model, yet it's slightly confusing to insert insulin resistance solely as a first hit. Consider being more inclusive about possible first hit components, such as sedentary lifestyle, high-fat diet, and obesity, as mentioned in the cited paper: "Hepatic accumulation of lipids secondary to sedentary lifestyle, high fat diet, obesity, and insulin resistance, acts as the first hit, sensitizing the liver to further insults acting as a ‘second hit’" (Buzzetti et al., 2016). Highlight the fact that while insulin resistance is vital, it's not the exclusive "first hit" for NAFLD. Mention that there's literature showcasing NAFLD cases without insulin resistance (see https://www.mdpi.com/1422-0067/17/5/633).

We agree with reviewers’ opinion about the "two-hit" model proposed for NAFLD development. Moreover, we mentioned into the manuscript and in the previous revision that NAFLD presents a multifactorial character sustained by the numerous molecular implications in the development and progression. In addition, we added more data about the proposed components of NAFLD first hit referring to sedentary lifestyle, high fat diet, obesity, and insulin resistance: ”In the category of first hit for NAFLD are included a series of risk factors such as: overnutrition, obesity, dietary composition, and sedentary lifestyle, along with the well-known insulin resistance (IR) [53]. There are several NAFLD cases that are not linked with to IR as the main factor in its development [51]. Overnutrition and dietary composition patterns are usually a mark of obesity, which is more pronounced when it is associated with an inactive lifestyle. For example, a recent meta-analysis showed that the prevalence of NAFLD in over-weight and obese people is almost 70%, recording the highest prevalence for America’s area [54]. On the other hand, China records a 30% prevalence rate for NAFLD prevalence, but the rate of it among obese children / adolescents was comprised between  57.6 – 71.7% for different regions of the country [55]. Closely related to this, the lack of physical activity has been correlated with the prevalence of NAFLD [56], [57]. It seems that technological advances have led to the disappearance of healthy habits and the promotion of sitting activities. A study from 2020 and another one from 2023 that evaluated the South Korean population proved that there is a positive correlation between NAFLD and time spent sitting. Moreover, it has emerged that the risk of developing NAFLD increases in direct proportion to the time spent inactivity or sitting [58], [59]. As mentioned above, the dietary composition also contributes to the occurrence, progression, and support of NAFLD-specific molecular changes. In the context of general NAFLD, it is thought that a high-fat diet heavy in trans fats, saturated fats, cholesterol, and fructose-containing beverages may raise visceral fat, enhance lipid accumulation in the liver, and accelerate the progression to steatohepatitis and fibrosis [60], [61]. Although obese individuals manifest a higher incidence of NAFLD, this trend starts to change  for non-obese individuals, according to recent data [62], [63].” and ”In some cases, the first hit of NAFLD is not represented by the IR [66], [67]. The existence of several genetic variants that appear as single nucleotide polymorphisms (SNP) suggests multiple genetic pathways for NAFLD. The most well-known SNP for NAFLD is rs738409 C>G which encodes the protein variant of Patatin-like phospholipase do-main-containing 3 (PNPLA3) [68], [69]. This promotes the fast and aggressive NAFLD progression to NASH via altered lipid catabolism within the hepatocytes containing endoplasmic reticulum and lipid vesicles membranes presenting the PNPLA3 I148M variant [70]. The effect of genetic variants in the PNPLA3 gene on NAFLD can differ between ethnic groups; for instance, the Chinese people having the lowest effect compared to Indians or Malays [71]. A recent genome-wide association study also identified several new predisposition locations for genes implicated in steroid dehydrogenase oxidoreductase activity modulation (17-beta hydroxysteroid dehydrogenase 13 - HSD17B13) and endo-plasmic reticulum membrane-modulated lipid metabolic processes (Transmembrane 6 superfamily Member 2 - TM6SF2) [72], [73]. HSD17B13 is a protein found on the surface of the lipid droplets within hepatocytes, whose loss was associated with a reduction in NAFLD progression to NASH or cirrhosis [74]. The TM6SF2 was recently reported as an independent risk factor for NAFLD and used as predictor for fibrosis progression and HCC occurrence [75].

  1. Despite the figure legend, Figure 2 still requires modification. Replace "disturbance" with "sustain the progression."

As it was indicated, we replaced the word ”disturbance” with ”sustain the progression”. Also, we added several improvements of the figure composition by placing new information that is linked with the figure’s legend.

  1. Adding a figure illustrating the relationship between coagulation factors and NAFLD will significantly enhance the paper, as it aligns with one of the central aims of the article.

The figure 2 was converted to figure 3. The new figure 2 is depicting the possible interactions between coagulation factors and NAFLD that are established when the context is present; this is added following the reviewer’s suggestion.

Thank you!

Kind regards,

The authors.
